# Investigation of the Effects of Polymer Dispersants on Dispersion of GO Nanosheets in Cement Composites and Relative Microstructures/Performances

**DOI:** 10.3390/nano8120964

**Published:** 2018-11-22

**Authors:** Shenghua Lv, Haoyan Hu, Yonggang Hou, Ying Lei, Li Sun, Jia Zhang, Leipeng Liu

**Affiliations:** 1College of Bioresources Chemical and Materials Engineering, Shaanxi University of Science and Technology, Xi’an 710021, China; Hyhy0609@outlook.com (H.H.); Houyonggang@sohu.com (Y.H.); leiying0623@sohu.com (Y.L.); sunli0530@sohu.com (L.S.); zhangjia201611@yahoo.com (J.Z.); Liuleipeng@sust.edu.cn (L.L.); 2School of Materials Science and Chemical Engineering, Xi’An Technologyical University, Xi’an 710021, China

**Keywords:** graphene oxide, nanosheets, polymer dispersants, intercalation composites, cement composites

## Abstract

This study focused on the uniform distribution of graphene oxide (GO) nanosheets in cement composites and their effect on microstructure and performance. For this, three polymer dispersants with different level of polar groups (weak, mild, and strong) poly(acrylamide-methacrylic acid) (PAM), poly(acrylonitrile-hydroxyethyl acrylate) (PAH), and poly(allylamine-acrylamide) (PAA) were used to form intercalation composites with GO nanosheets. The results indicated that GO nanosheets can exist as individual 1–2, 2–5, and 3–8 layers in GO/PAA, GO/PAH, and GO/PAM intercalation composites, respectively. The few-layered (1–2 layers) GO can be uniformly distributed in cement composites and promote the formation of regular-shaped crystals and a compact microstructure. The compressive strengths of the blank, control, GO/PAM, GO/PAH, and GO/PAA cement composites were 55.72, 78.31, 89.75, 116.82, and 128.32 MPa, respectively. Their increase ratios relative to the blank sample were 40.54%, 61.07%, 109.66%, and 130.29%, respectively. Their corresponding flexural strengths were 7.53, 10.85, 12.35, 15.97, and 17.68 MPa, respectively, which correspond to improvements of 44.09%, 64.01%, 112.09%, and 134.79%.

## 1. Introduction

Graphene oxide (GO), a derivative of graphene, not only inherits the structural and property-related features of graphene such as a large surface area and high strength, but also exhibits unique properties such as the presence of a large number of oxygen-rich functional groups and good hydrophilicit [1,2] The unique structure and properties of GO give it potential for use in improving the mechanical properties and durability of cement composites. Recent research has shown that GO can be used to improve cement composite performance. This has become an active area of research and received extensive attention [3,4]. Several studies have investigated the effects of dosage [5], chemical groups [6,7], and the GO nanosheet size range [8,9] on the rheological properties [10,11,12], hydration reactions and hydration products [13,14,15], microstructures [16,17], mechanical properties [18,19], and durability [20] of cement composites, as well as mechanisms of action [21]. The common conclusion is that addition of GO nanosheets to cement composites can improve the mechanical strength and regular microstructure of the latter materials [22,23,24,25]. A mechanism has been proposed which suggests that GO nanosheets not only play a filling, interlocking, and bridging function between cracks and pores within cement composites, but can also enhance the regularity of crystalline reaction products and compactness of the microstructure, resulting in reinforcing and toughening effects [26,27].

Cement composite is a type of porous, brittle material [28]. Normal cement composites consist mainly of an amorphous solid with a complex shape. The traditional understanding is that the porosity stems primarily from the free water in cement hydration products and the interfacial gaps. However, the pores are shielded and allow no permeation, so this effect is limited. Research has found that the interfacial gaps consisting mainly of large numbers of communicating cracks and pores can significantly diminish their performance [29]. The interfacial porosity and cracks within normal cement composites are caused primarily by the complex shapes of cement hydration products, resulting in the production of many gaps, cracks, and pores in cement composites with diameters in a range of 10 nm–1.0 mm, and they cannot be uniformly distributed in the cement matrix [29]. Research on decreasing the porosity and cracks has been performed, but this problem is difficult to solve using traditional methods.

Research has discovered that GO nanosheets can control the shapes of cement hydration products, causing them to form compact, regular microstructures, indicating a potential approach to solving this problem [30,31]. Unfortunately, it is difficult to distribute GO nanosheets uniformly in cement composites as individual few-layered nanosheets. As a result, the cement hydration crystals often distribute and grow unevenly in the cement composites, forming large interfacial gaps and cracks. This occurs primarily because GO nanosheets aggregate easily, re-stacking in suspensions and cement composites due to large-area π–π interactions and strong Van der Waals interactions between graphene oxide layers [32]. Currently, this problem is addressed primarily by adding only a low dosage of GO nanosheets during GO/cement composite preparation [33,34]. In addition, ultrasonication and dispersants are commonly used to disperse the nanosheets [35,36].

In our previous study, we also determined and confirmed that GO nanosheets can force cement hydration products to form regularly shaped hydration products and build them into compact, uniform microstructures by intercalating and crosslinking the crystals. This reinforces the cement composites [26,34]. The reinforcing effects of GO are closely related to nanosheet dispersion. As mentioned earlier, even distribution of GO nanosheets in cement composites is difficult to achieve due to strong interactions between the layers. This results in an uneven distribution of hydration crystals in the cement composites and the production of more interfacial gaps and cracks in the composite structure. Therefore, we studied the effects of dispersants of polycarboxylate superplasticizers (PCs), polyacrylate, and carboxymethyl chitosan on GO nanosheet distribution by forming intercalating composites or grafting copolymers in previous research [17,24,26,34,37,38]. The results indicated that the PCs and polyacrylate cannot uniformly distribute GO in cement composites with few layers by forming composites due to their poor intercalation with GO nanosheets, and the resulting cement composites have an uneven and porous microstructure. Meanwhile, the results also indicated that forming grafting copolymers of PCs and polyacrylate with GO would obviously decrease the number of active groups on the nanosheets and, therefore, their ability to control the cement microstructure [36]. Carboxymethyl chitosan has excellent dispersibility for GO nanosheets and can disperse the multilayered GO agglomeration into few-layered GO nanosheets (1–2 layers) due to its amphoteric polymeric structure, but it has a high price and is not appropriate for widespread use. However, the amphoteric carboxymethyl chitosan enlightens us to the possible synthesis of a polymeric amphoteric dispersant to replace carboxymethyl chitosan.

Based on the above results and analysis, we think that the application of suitable dispersants may solve the agglomeration problem of GO nanosheets in cement composites. The dispersive character of a dispersant depends primarily on its structure, including its volume, polarity, and charge characteristics. In this paper, we select three polymer dispersants with different polar groups (weak, mild, and strong) and investigate their ability to disperse GO nanosheets. Our goal is to prepare GO nanosheets made up of only a few layers and uniformly distribute them in an aqueous suspension solution by forming GO/dispersant intercalation composites. Then, through doping the GO/dispersant intercalation composites in cement composites, we aim to distribute GO nanosheets uniformly and individually in cement composites to form compact, uniform microstructures throughout the materials, improving their performance. The results serve to reveal the effects of polymer dispersants with differing polar groups on GO nanosheets and correlate them to cement hydration products, interfacial gaps, microstructure, and performance. This research provides a pathway to the preparation of high-performance cement composites with compact microstructures by incorporating synthesized amphoteric polymer dispersants.

## 2. Experimental Section

### 2.1. Materials

Ordinary Portland cement 42.5 (P.O.42.5) was produced by Shengwei Cement Co. Ltd. (Xi’an, China). The original suspension solution of GO sheets was produced using a modified Hummers method [39]. The GO content was 0.5% and the ranges of planar size and thickness were 110–560 nm and 10.5–17.8 nm, respectively. The polymer dispersants poly(acrylamide-methacrylic acid) (PAM), poly(acrylonitrile-hydroxyethyl acrylate) (PAH), and poly(allylamine-acrylamide) (PAA) were synthesized via free-radical copolymerization with a monomer molar ratio of 1:3 according to order of polymer name. Their average molecular weights (*M*_w_, *M*_n_) and polydispersity (PDI), as well as dielectric constants, are shown in Table 1. Their chemical structures are shown in Figure 1. Polycarboxylate superplasticizers (PCs) are the most common cement additive and were supplied by Kezhijie Additive Co. Ltd. (Xi’an, China). Its water-reducing ratio is 33.5% and its function is to reduce the mixing water dosage and improve the fluidity of fresh cement composites.

### 2.2. Preparation of GO/PAM, GO/PAH, and GO/PAA Intercalation Composites

GO/PAM, GO/PAH, and GO/PAA intercalation composites were prepared via ultrasonic treatment of a mixture of 100 g of a 0.5% GO sheet suspension, 130 g of water, and 20 g of a 10% PAM, or PAH, or PAA solution for 60 min. The goal was to prepare few-layered GO nanosheets with the potential to be uniformly distributed throughout the composites as individual nanosheets.

### 2.3. Preparation of GO/Cement Composites

The GO/cement composites were prepared by mixing cement, water, PCs, and GO at a weight ratio of 100:25:0.30:0.03. The PCs and GO dosages described in the recipe were solid dosages by cement weight, while the water dosage included the water from the GO intercalation composite and PCs solution. The GO dosage of 0.03% was optimized based on previous results [25]. The blank sample consisted of free GO nanosheets. A control sample was prepared by directly adding original GO nanosheets. GO/PAM/cement, GO/PAH/cement, and GO/PAA/cement composites were prepared by doping GO/PAM, GO/PAH, and GO/PAA intercalation composites, respectively, via the same procedure. For convenience of analysis and discussion, S_1_, S_2_, S_3_, S_4_, and S_5_ are used to represent the blank and control samples, and the GO/PAM/cement, GO/PAH/cement, and GO/PAA/cement composites, respectively.

### 2.4. Structural Characterization Methods

The chemical structures, surface morphologies, and size distributions of the GO nanosheets in the intercalation composites were characterized via a Fourier transform infrared spectroscopy (FTIR; EQUINOX-55, Bruker, Ettlingen, Germany) and a X-ray photoelectron spectroscope (XPS; XSAM 800, Kratos, Manchester, UK).

The surface morphologies of the GO nanosheets were characterized using a atomic force microscope (AFM; SPI3800N/SPA400, Seiko, Osaka, Japan) and a transmission electron microscope (TEM; Tecnai G2F20, FEI, Hillsboro, OR, USA). A purified sample (0.5% concentration) was diluted 1000-fold. A drop of the resulting solution was placed on a monocrystalline silicon wafer and dried in a vacuum oven (50 °C, 1 h) for AFM imaging. The samples used for TEM were placed on copper grids and dried naturally. The size distribution of the GO nanosheets in suspension was determined using a laser particle analyzer (LPA; NANO-ZS90, Zetasizer, Worcestershire, UK). The XRD patterns of the GO nanosheets were obtained by using a X-ray diffraction tester (XRD; Rigaku D/max2200PC, Rigaku, Osaka, Japan) with Cu Kα radiation to measure freeze-dried GO nanosheets. XRD patterns can be used to characterize the order and layer spacing of GO.

The microstructures of the cement composites were characterized using a scanning electron microscope (SEM; S-4800, Hitachi, Tokyo, Japan). The elemental contents of selected areas were mapped using an EDAX energy-dispersive X-ray spectrometer (EDAX, Cassatt, South Carolina, USA). The XRD patterns of the cement composites were confirmed by using the tool described above to identify the crystal phases of the cement composites. Pore structures were measured using an automatic mercury porosimeter (AMP; Autopore IV9500, Micromeritics, Norcross, GA, USA).

### 2.5. Performance Testing Methods

The compressive and flexural strengths of the cement composites were tested using a concrete compressive strength tester (JES-300, Jianyi, Wuxi, China) that increased the pressure at a rate of 2.4–2.6 MPa/s and a concrete three-point flexural strength tester (DKZ-500, Jianyi, Wuxi, China) with a pressure increase rate of 1 MPa/s. For each recipe, five samples were measured and the results averaged. The samples were molded with a size of 40 mm × 40 mm × 160 mm to test the flexural and compressive strength.

The durability indices include water penetration depth, freeze–thaw mass loss, and carbonation depth. These were measured according to the China National Standard GB/T5082-2009. These specimens for testing water penetration depth, freeze–thaw mass loss, and carbonation depth were molded as a cylinder with a size of φ150 mm × 150 mm, a cuboid with a size of 110 mm × 110 mm × 500 mm, and a cube with a size of 100 mm × 100 mm × 100 mm, respectively.

## 3. Results and Discussion

### 3.1. Structural Characterization of GO Nanosheets

The structures of the GO nanosheets were characterized via FTIR and XPS, and the results are shown in Figure 2. Figure 2a shows the FTIR spectrum of GO nanosheets. The spectrum indicates the presence of hydroxyl (–OH, 3353 cm^−1^), carboxyl (COOH, 1731 cm^−1^), carbonyl (C=O, 1620 cm^−1^), and ether (–C–O–C–, 1420, 1360, 1250, 1130 and 1060 cm^−1^) groups. Figure 2b shows an XPS spectrum of GO, which suggests that C=C/C–C, C–O–C, C=O, and COOH are present in the following respective relative quantities: 31.67%, 33.56%, 15.26%, and 19.51%.

The micromorphologies of GO nanosheets including their interlayer spacings and size distributions were obtained via XRD and LPA. Figure 2c shows the XRD patterns of graphite and GO. It shows that the interlayer spacings of graphite, unmodified GO, GO/PAM, GO/PAH, and GO/PAA are 0.35 nm, 0.73 nm, 0.75 nm, 0.78 nm, and 0.85 nm, respectively. The results suggest that GO sheets have larger interlayer spacings when they are in intercalation composites. Figure 2d shows the size distributions of GO nanosheets as determined via LPA. The size ranges detected in the original GO suspension and the GO/PAM, GO/PAH, and GO/PAA composites are 20–950 nm, 4–410 nm, 2–290 nm, and 1–110 nm, respectively. Thus, the dispersants can be ordered by dispersive capacity (from weak to strong) as follows: PAM, PAH, PAA. This is consistent with the order of dispersant polarities and dielectric constants.

The surface morphologies were obtained via TEM. TEM images of the GO nanosheets are shown in Figure 3. They serve to express the dispersion of the GO nanosheets qualitatively. GO sheets made from GO/PAH (see Figure 3c) and GO/PAA (see Figure 3d) have smaller planar sizes and are more unlikely to re-stack than those made from GO/PAM (see Figure 3b). Sizes decrease in the following order: PAM, PAH, PAA. The planar size is approximately 300–800 nm, and GO nanosheets can exist as few-layered sheets that are uniformly distributed in GO/PAH and GO/PAA composites (see Figure 3c,d). This may occur because PAH and PAA are stronger polar polymer dispersants than PAM. They easily adsorb GO nanosheets, avoiding re-stacking via co-action of steric hindrance and electrostatic repulsion.

AFM images of GO nanosheets can be used to quantitatively characterize their planar sizes and thicknesses. Figure 4 shows AFM images of the original GO sheets. Figure 4a,b indicate the top view and side view of original GO sheets, respectively. The results indicate that the GO sheets aggregate easily, and their planar size range is 210–950 nm. Figure 4a_1_–a_4_ show the side view of GO sheets in a_1_, a_2_, a_3_, and a_4_ in Figure 4a, indicating that these GO sheets aggregation are rough and uneven, and their thickness range is 12.5–19.6 nm (8–15 layers). Figure 5 shows the AFM images of GO sheets form the intercalation composites. Figure 5a shows AFM images of GO sheets made from GO/PAM composites. The planar sizes range from 120 nm to 550 nm, and their sheet thicknesses are less than 7.67 nm. The profile view of the GO nanosheets indicates thicknesses of about 3.5–4.5 nm (see Figure 5a_1_,a_2_), which suggests that they are composed of 4–8-layer stacks. Individual GO nanosheets are flatter than GO nanosheets made up of several layers. Figure 5b shows an AFM image of GO sheets from GO/PAH composites. The GO nanosheets are small at 100–350 nm and are 4.32 nm thick. The profile view indicates that GO nanosheets that consist of 2–4 layers and exhibit thicknesses of 1.9–2.3 nm can be individually distributed in suspension (see Figure 5b_1_,b_2_). Figure 5c shows an AFM image of GO sheets from GO/PAA composites. These GO sheets exhibit the best level of dispersion. Their planar size range is 30–130 nm and they are less than 3.26 nm thick. The results suggest that GO nanosheets exist primarily as few-layered nanosheets (profile thickness: 0.9–1.1 nm, 1–2 layers, see Figure 5c_1_,c_2_) and can be uniformly and individually distributed in the intercalation composites. The results indicate that PAA has greater dispersive capacity than the other dispersants tested.

Based on the results of the size distribution, AFM images, and TEM images of GO nanosheets, a schematic diagram of the dispersing mechanism of dispersants for GO sheets is proposed and shown in Figure 6. Figure 6a shows that original GO undergoes multilayer aggregation. There are processes of dispersion and restacking in the GO suspension. The restacking capacity of the GO nanosheets is stronger than its dispersing. Therefore, the GO suspension usually exhibited multilayer agglomeration (>5 layers). Figure 6b shows the dispersing mechanism of PAA for the GO nanosheets in the GO suspension. PAA has the strongest polar groups (–NH_2_) of the three dispersants. So, the interactions between the PAA and the polar groups of GO nanosheets are major ionic bonding and electrostatic and Van der Waals forces in the GO/PAA intercalation composites. The stability of GO/PAA composites is stronger and results in forming a few-layered (1–2 layer) GO dispersion (Figure 6b). In particular, the PAA has strongest polar group of –NH_2_ compared to PAM and PAH, and the –NH_2_ will be protonated to form NH_3_^+^ in the neutral or weak acidic solution of the mixture of PAA and GO. This will produce stronger ionic bonding between PAA and GO nanosheets by the –NH_3_^+^ of PAA and the –COO of GO in a neutral or weakly acidic mixing dispersion, resulting in the formation of a few-layered (1–2 layers) and uniform GO dispersion. When the GO/PAA composites are added into cement composites, the environment will change to alkaline (pH 8–10), the –NH_3_^+^ will be gradually deprotonated, and the ionic bonding between PAA and GO will also decrease and disappear; finally, the free GO will be released to control cement reaction products (Figure 6b). In these processes, GO nanosheets have been dispersed uniformly in cement composites. The results that the NH_2_ groups are certainly protonated in solutions of neutral media forming –NH_3_+ groups but are deprotonated in alkaline solutions have been analyzed by other researchers [40,41,42,43]. The dispersant effects were attributed to synergies between steric hindrance and electrostatic repulsion.

### 3.2. Cement Composite Microstructure

The microstructures of cement composites can be characterized using SEM images and can be evaluated for compactness, uniformity, and porosity. Before the effect of few-layered GO nanosheets on the cement composite microstructures was investigated, the microstructures of S_1_ and S_2_ were viewed via SEM at 28 day after preparation. The results are shown in Figure 7. Figure 7a,b show large numbers of pores and cracks in S_1_ due to the presence of irregular cement hydration products. There are small quantities of needle-like and sheet-like products in S_1_. These are stacked in a disorderly manner and produce various interfacial gaps and cracks. Figure 7c,d show the microstructure of S_2_, which is doped with a suspension of unmodified GO nanosheets. Various aggregated cement hydration crystals are distributed in the composites. They form interleaved structures within the aggregations but do not form large-scale compact structures that spread across the composites. This result suggests that the original GO nanosheets are unevenly distributed in the cement composites.

Figure 8 shows SEM images of S_3_ at 28 day. Figure 8a indicates that there are large numbers of aggregated crystals in the cement composite and suggests that GO nanosheets are nonuniformly distributed. Figure 8b shows clearly that cement hydration crystals form separate, individual aggregates. Although the crystalline aggregates nearly form large-scale compact structures, they are too far apart to achieve this. Figure 8c shows that the larger-volume aggregates consist of several interweaved, regularly shaped nanocrystals. The unevenly distributed crystals produce various interfacial pores and cracks. Figure 8d shows a carbon map of the area presented in Figure 8b. It suggests that GO nanosheets are unevenly distributed in the cement composites. These experimental results indicate that PAM can disperse GO nanosheets only into groups 3–8 layers thick and cannot distribute them evenly in the cement composites. Although large numbers of regularly shaped crystals can be produced via doping with GO nanosheets, they exist as isolated, large-volume aggregates and are unevenly distributed in the cement composites.

Figure 9 shows SEM images of S_4_ at 28 day. Figure 9a–c indicate that the cement composites have compact, regular microstructures that consist of regularly shaped nanocrystals in flower-like patterns. The results suggest that these regularly shaped nanocrystals are produced uniformly throughout the composites and can interweave and crosslink to form compact microstructures. The interfacial gaps and cracks between crystals and aggregations have disappeared. Figure 9d shows a carbon map of the area displayed in Figure 9b. It indicates that carbon is more uniformly distributed in S_4_ than in S_2_. Thus, Figure 9 indicates that PAH has a strong dispersive capacity and not only disperses GO into 2–5-layer nanosheets but also distributes them uniformly throughout the cement composites. The resulting microstructure can be attributed to the presence of GO nanosheets in groups of only a few layers and their uniform distribution in the composites.

Figure 10 shows SEM images of S_5_. Figure 10a–c show compact, regular microstructures throughout the composites. The structures consist of regular, interweaved polyhedron-like crystals. They exhibit more compactness and uniformity than those in S_4_, which indicates that the crystalline products grow uniformly in the cement composites. This occurs because PAA has a strong dispersive capacity and can disperse GO into 1–2-layer nanosheets that are evenly distributed throughout the composites. The microstructure does not exhibit interfacial gaps or pores/cracks. Figure 10d shows a carbon map of the area indicated in Figure 10b. Carbon is distributed evenly throughout the area. Thus, PAA can disperse GO nanosheets into groups of a few layers and evenly distribute them in cement composites to form a compact microstructure.

The above results indicate that S_4_ and S_5_ exhibit more compact, uniform microstructures than S_2_ and S_3_. Thus, the dispersants can be ordered by dispersive capacity (from weak to strong) as follows: PAM, PAH, PAA.

### 3.3. Forming Process of Regular Hydration Crystals and Compact Microstructures

The SEM images discussed above indicate that introduction of GO nanosheets can cause cement hydration products to form regularly shaped crystals that aggregate into compact microstructures at 28 day. The cement hydration reaction and microstructure formation take about 28 days. Therefore, observing the microstructures at intermediate ages such as 1 day, 3 day, 7 day, and 15 day may help one to understand the associated formation mechanisms. Figure 11 shows SEM images of cement composites at 1 day, 3 day, 7 day, 15 day, and 28 day.

Figure 11a_1_–a_5_ show the microstructure of S_3_ at various hydration ages. The initial crystals are generated at 1–3 days and grow from 3 day to 7 day. The finished microstructure is presented from 7–15 days, and a perfect compact microstructure is formed at 28 day. The most significant feature is that many large-volume aggregates of regularly shaped crystals are unevenly and separately distributed in the composites, resulting in a lack of compactness.

Figure 11b_1_–b_5_ show SEM images of S_4_ at various hydration ages. Large numbers of bar-like and flower-like crystal products are present at 1 day. They grow to form flower-like patterns at 3 day, with microstructural crosslinking nearly finished at 7–15 days. The final compact, ordered microstructure is formed at 28 day. The results indicate that the composites must undergo initial crystal production and growth before interweaving to form a compact microstructure.

Figure 11c_1_–c_5_ show the microstructure of S_5_ at various hydration ages. The most outstanding feature of this system is that its microstructure is more compact than those of the composites discussed previously. Figure 11c_1_ shows that large-scale crystals with interfacial gaps are generated after 1 day. Figure 11c_2_ shows that the compact crystal aggregates begin to form at 3 day. Figure 11c_3_ shows that the compact, ordered microstructure is nearly complete at 7 day. Figure 11c_4_ shows that the compact microstructure is almost fully formed at 15 day. Figure 11c_5_ shows a final, perfect compact microstructure at 28 day. The results in Figure 11c_1_–c_6_ further suggest that uniformly and individually distributed few-layered GO nanosheets are the key to preparing cement composites with ordered, uniform, compact microstructures.

The results discussed above demonstrate that uniform distribution of few-layered GO nanosheets in cement composites is the key to achieving the desired microstructure. Introducing GO/PAH and GO/PAA intercalation composites into cement composites can result in cement composites with uniformly dispersed GO nanosheets. The GO surface contains oxygen-rich chemical groups that become templates for growth of the initial hydration crystals at 1 day. The directions of crystal growth are limited by the orientations of the GO nanosheets. Thus, the crystals tend to form ordered microstructures via crosslinking at 3–7 days. The final ordered microstructure is completed at 28 day.

### 3.4. Relationship of Interface Gaps of Hydration Crystals with the Hydration Reaction Process

The above results indicate that GO-nanosheet-induced formation of ordered microstructures takes place gradually. The outstanding structural feature is that there is greater porosity in the early days of hydration ages (1–7 days). The porosity includes pores and cracks that form which are mostly interfacial gaps between crystalline reaction products. These interfacial gaps may be characterized by measuring the porosities of cement composites. The pore structure test results are shown in Figure 12. The interfacial areas and pore diameters of cement composites are shown in Table 2.

Figure 12a shows the relationship between the interfacial areas within the cement composites and the cement hydration age. The interfacial areas of all cement composites decrease as the hydration time increases from 1 day to 28 days. The interfacial areas of S_1_ and S_2_, at 1 day/3 day/7 day/15 day/28 day are 25.68/20.41/17.57/16.52/15.68 m^2^/g and 23.65/18.78/13.67/11.25/10.65 m^2^/g, respectively. The interfacial areas of S_3_ are similar to those of S_2_. In contrast, the interfacial areas of S_4_, and S_5_ at the same hydration ages are 21.08/12.06/4.15/4.74/4.25 m^2^/g and 18.97/10.03/3.14/3.05/2.92 m^2^/g. The results indicate that S_4_ and S_5_ clearly exhibit smaller interfacial areas than S_1_, S_2_, and S_3_ at 7 day, 15 day, and 28 day. Meanwhile, this occurs because all cement hydration products become regular crystals in S_4_ and S_5_. The initial crystals are produced at 1 day and grow quickly between 3 day and 7 day. Thus, the interfacial area decreases significantly from 1 day to 7 day. The compact microstructure is nearly formed at 15 day and is complete at 28 day. The volumes of the growing crystals increase to fill the gaps between crystal interfaces, resulting in a compact, crosslinked microstructure. Figure 12b shows the pore diameters of cement composites at different hydration ages. S_4_ and S_5_ have large pores at 1 day. The pore sizes decrease significantly between 3 day and 7 days. Finally, the pore diameter reaches 11–14 nm at 28 day. In contrast, S_1_, S_2_, and S_3_ exhibit large pore diameters at 1 day, followed by small decreases in pore size between 3 day and 7 day. The cured materials exhibit pore diameters of 33–47 nm, which are significantly larger than those seen in S_4_ and S_5_.

Based on the above analysis, one can conclude that porosity in cement composites is caused primarily by interfacial gaps between crystals. These interfacial gaps are mostly correlated with the regularity of hydration products and their distribution in the cement composites.

### 3.5. XRD Pattern Analysis of Cement Hydration Products

The cement hydration products formed under the influence of GO and became regular crystals whose phases may be characterized via XRD. The XRD patterns and crystal chemical structures are shown in Figure 13. Figure 13a is the XRD pattern of S_1_, which is a blank sample made without GO nanosheet doping. The results indicated that the height of the XRD peak is smallest among the five XRD patterns, which suggested that crystal products are few, and the crystal products are CH, AFt, and AFm, and they exhibit hexagonal, tetragonal, and orthorhombic crystal types, respectively. The cement hydration products are composed of traditional amorphous hydration products and small amounts of tetrahedral, cubic, orthorhombic, monoclinic, and hexagonal crystal phases. Figure 13b–e are the XRD patterns of S_2_, S_3_, S_4_, and S_5_, and the results indicate that the heights of XRD peaks increase significantly from S_2_, to S_3_, to S_4_ to S_5_ in turn, which indicates the presence of more crystals in the cement composites of S_4_ and S_5_. The XRD patterns of S_5_ show that S_5_ has the strongest XRD pattern of all the samples, which suggests the presence of more crystal products, including amorphous C–S–H (Ca_3_Si_2_O_7_) gel that has been transformed into a monoclinic crystal. The above results indicate that S_4_ and S_5_ contain more crystals than do S_2_ and S_3_. Thus, the GO nanosheets can promote the production of more hydration crystals when they are present as uniformly distributed, few-layered nanosheets.

### 3.6. Cement Composite Compressive/Flexural Strengths and Durabilities

Table 3 shows the compressive strengths of the cement composites, indicating that the compressive strengths of the cement composites increase from S_1_, to S_2_, to S_3_, to S_4_, to S_5_ in turn. S_5_ has the maximum compressive strength among these cement composites. S_4_ and S_5_ show a greater increase compared with S_2_ and S_3_. At 28 day, the compressive strengths of S_1_, S_2_, S_3_, S_4_, and S_5_ are 55.72, 78.31, 89.75, 116.82, and 128.32 MPa, respectively. S_2_, S_3_, S_4_, and S_5_ are 40.54%, 61.07%, 109.66%, and 130.29% stronger than S_1_, respectively.

Table 4 shows the flexural strengths of the five cement composites, indicating that they have a similar increasing tendency to the compressive strength. The flexural strength increases from S_1_, to S_2_, to S_3_, to S_4_, to S_5_ in turn. S_5_ has the maximum flexural strength among the five cement composites. At 28 day, the flexural strengths of S_1_, S_2_, S_3_, S_4_, and S_5_ are 7.53, 10.85, 12.35, 15.97, and 17.68 MPa, respectively. Relative to S_1_, these values correspond to improvements of 44.09%, 64.01%, 112.09%, and 134.79%, respectively. The results indicate that cement composites containing GO nanosheets are stronger than the blank and control samples. Meanwhile, the cement composites improve in the following sequence: original GO nanosheets, GO/PAM, GO/PAH, and GO/PAA. This suggests that the dispersion of GO nanosheets in the cement composites significantly influences the mechanical strength of the resulting materials.

The interfacial gaps between cement hydration product surfaces within cement composites are a major factor in cement performance, especially with regard to durability. The pore areas and diameters stem primarily from interfacial gap areas and sizes. The durability of cement composites is usually evaluated via the penetration parameters, which is associated with pores in cement composite structures. The parameters are composed of penetration resistance, freeze–thaw mass loss, and carbonation depth, which are shown in Table 5. The results indicate that the seepage height, freeze–thaw mass loss, retention rate of the relative dynamic elasticity modulus, and carbonation depth improve significantly relative to the blank and control samples. This is attributed to compact, highly ordered cement composite structures which consist of interweaved, crosslinked, and regularly shaped crystals, which clearly increase the interfacial gaps.

## 4. Conclusions

(1)GO nanosheets with 1–2, 2–5, and 3–8 layers were produced by using PAA, PAH, and PAM, respectively, as dispersants to form GO–dispersant intercalation composites. The order of dispersive capacities from poor to strong is PAM, PAH, PAA. PAA and PAH contain strong –NH_2_ and –CN polar groups that result in stronger dispersion effects.(2)Cement composites with compact and uniform microstructures can be prepared by introducing GO/PAA intercalation composites into cement composites. The microstructure is formed by gradually growing large-scale regular cement hydration crystals. The results indicated that PAA can disperse GO in cement composites with few layers and a uniform distribution.(3)The cement composites with GO/PAA have significantly improved compressive strength and flexural strength. The cement composites with 0.03% dosage of GO/PAA at 28 day show 130.29% and 134.79% improvements in compressive and flexural strengths, respectively, when compared to the control samples. The results indicate that cement composites with few-layered and uniformly distributed GO nanosheets have a significantly improved in microstructure, strengths, and durability.

## Figures and Tables

**Figure 1 nanomaterials-08-00964-f001:**
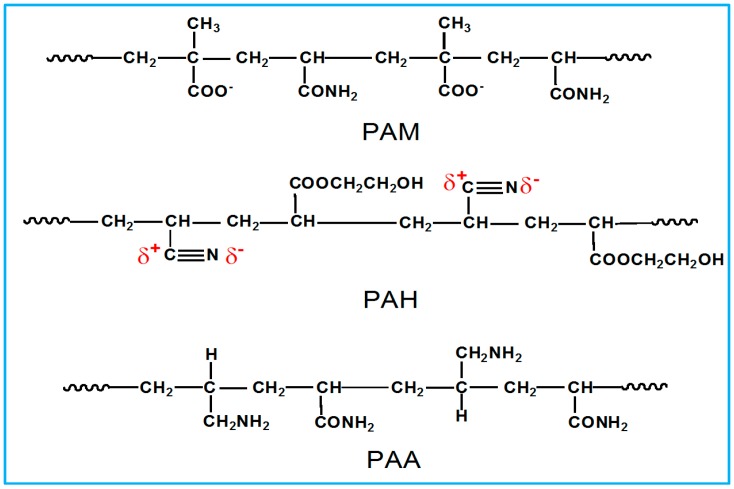
Schematic diagram of the chemical structures of PAM, PAH, and PAA.

**Figure 2 nanomaterials-08-00964-f002:**
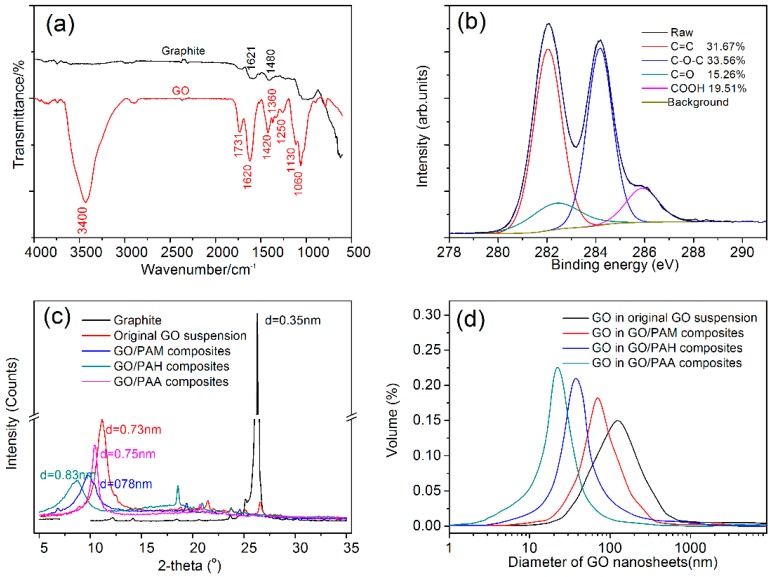
(**a**) FTIR spectra of graphene oxide (GO) and graphite. (**b**) XPS spectrum of GO. (**c**) XRD patterns and (**d**) size distributions of various GO nanosheets.

**Figure 3 nanomaterials-08-00964-f003:**
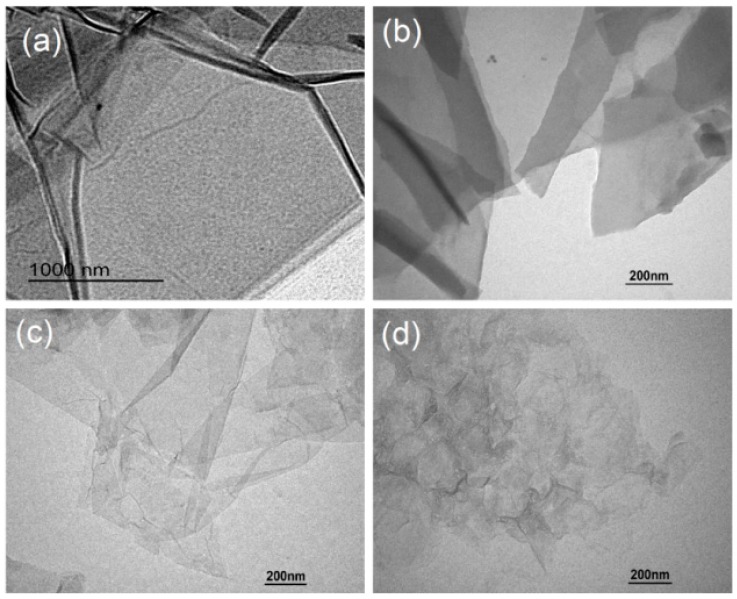
TEM images of GO nanosheets from (**a**) original GO suspension; (**b**) GO/PAM intercalation composites; (**c**) GO/PAH intercalation composites; and (**d**) GO/PAA intercalation composites.

**Figure 4 nanomaterials-08-00964-f004:**
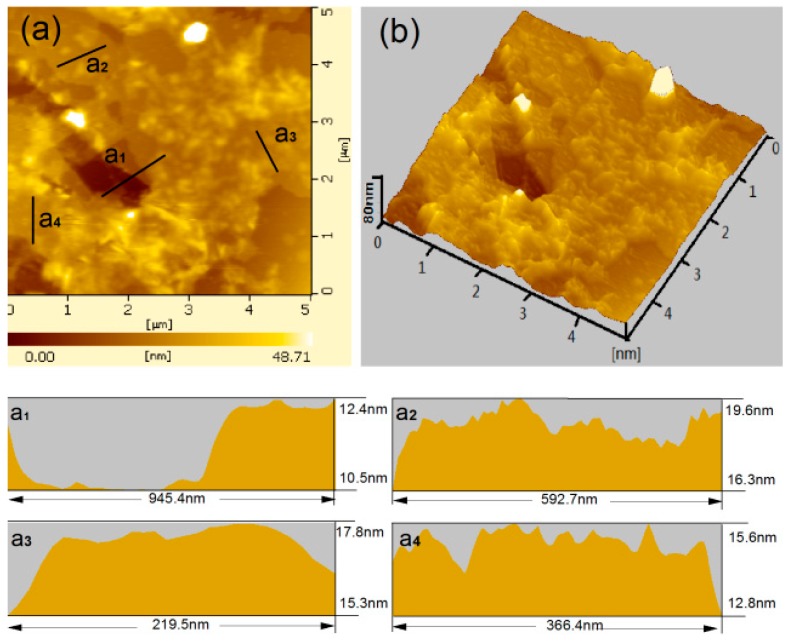
AFM images of GO sheets from the original GO suspension: (**a**) top view, (**b**) side view, (**a_1_**,**a_2_**,**a_3_**,**a_4_**) the side view of GO sheets at (**a_1_**,**a_2_**,**a_3_**_,_**a_4_**) in (**a**).

**Figure 5 nanomaterials-08-00964-f005:**
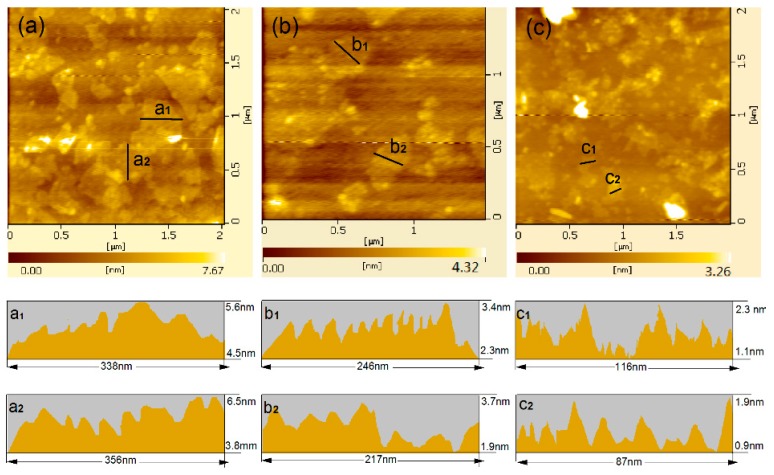
AFM images of GO nanosheets from (**a**) GO/PAM composites; (**b**) GO/PAH composites; and (**c**) GO/PAA composites. (**a_1_**,**a_2_**,**b_1_**,**b_2_**,**c_1_**,**c_2_**) are side views of GO at the corresponding lines.

**Figure 6 nanomaterials-08-00964-f006:**
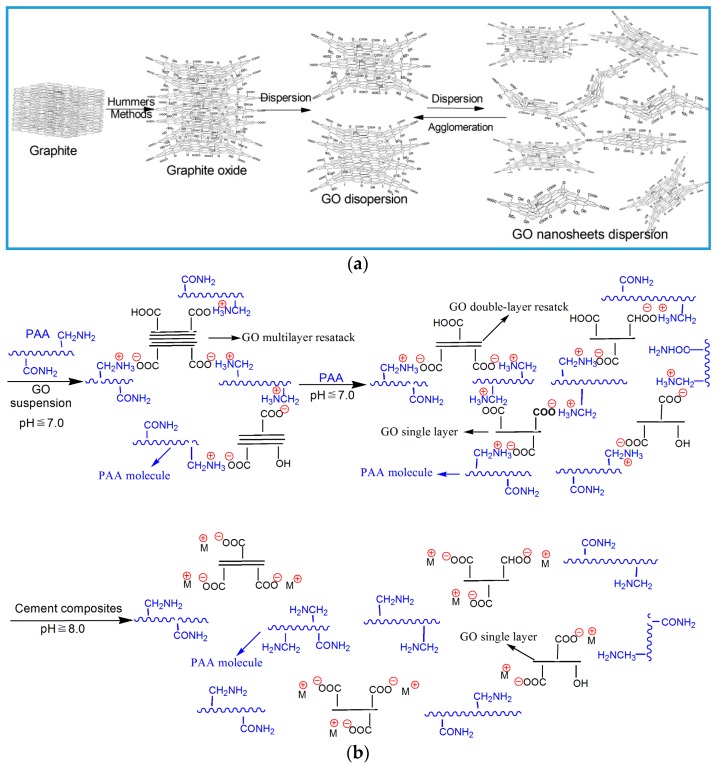
Schematic diagram of the dispersing mechanism of GO nanosheets with PAA. (**a**) Original GO nanosheet suspension; (**b**) Dispersing mechanism of GO nanosheets with PAA.

**Figure 7 nanomaterials-08-00964-f007:**
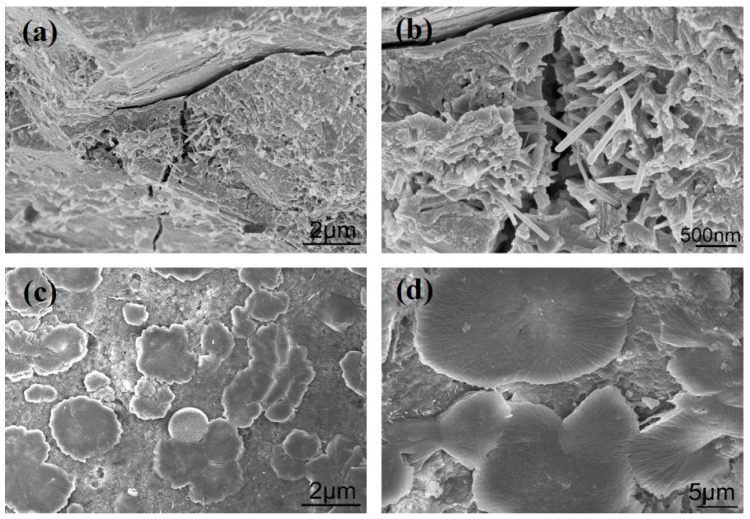
SEM images of S_1_ samples (**a**,**b**) and S_2_ samples (**c**,**d**).

**Figure 8 nanomaterials-08-00964-f008:**
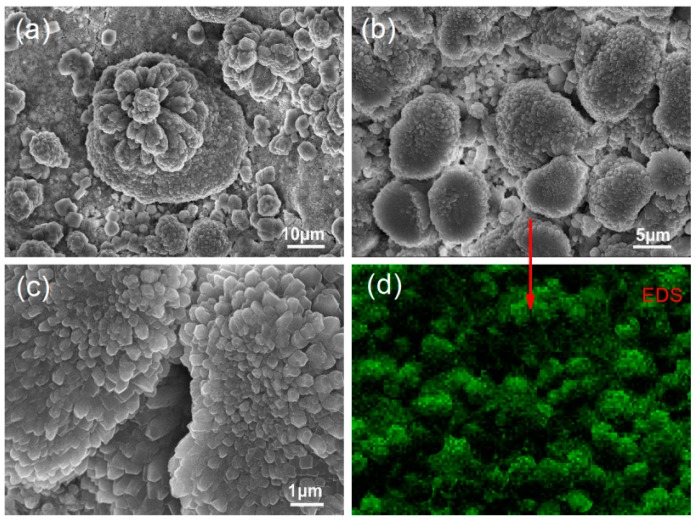
(**a**–**c**) SEM images of GO/PAM/cement composites with unmodified GO nanosheets. (**d**) Carbon map of the area presented in (**b**).

**Figure 9 nanomaterials-08-00964-f009:**
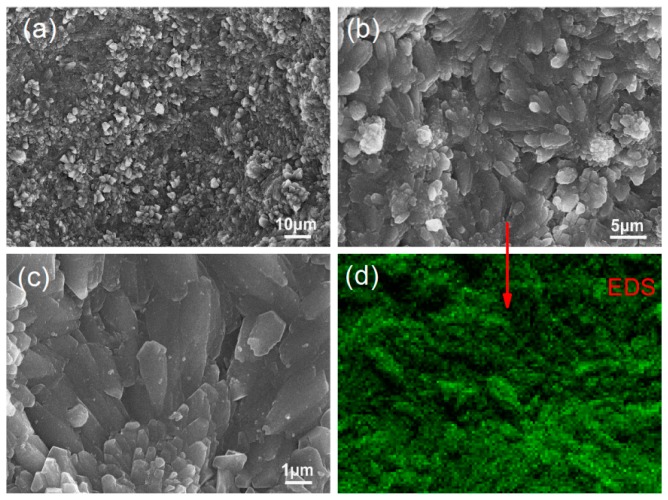
(**a**–**c**) SEM images of GO/PAH/cement composites with GO/PAH intercalation composites. (**d**) Carbon map of the area presented in (**b**).

**Figure 10 nanomaterials-08-00964-f010:**
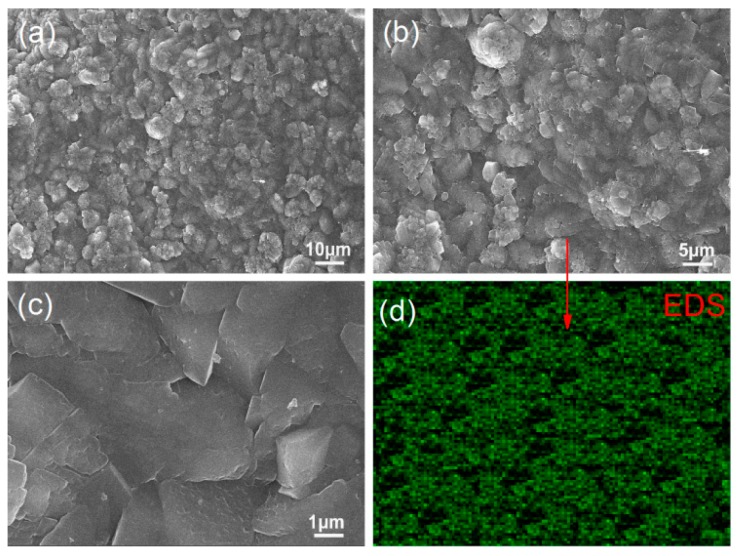
(**a**–**c**) SEM images of GO/PAA/cement composites with GO/PAA intercalation composites. (**d**) Carbon map of the area presented in (**b**).

**Figure 11 nanomaterials-08-00964-f011:**
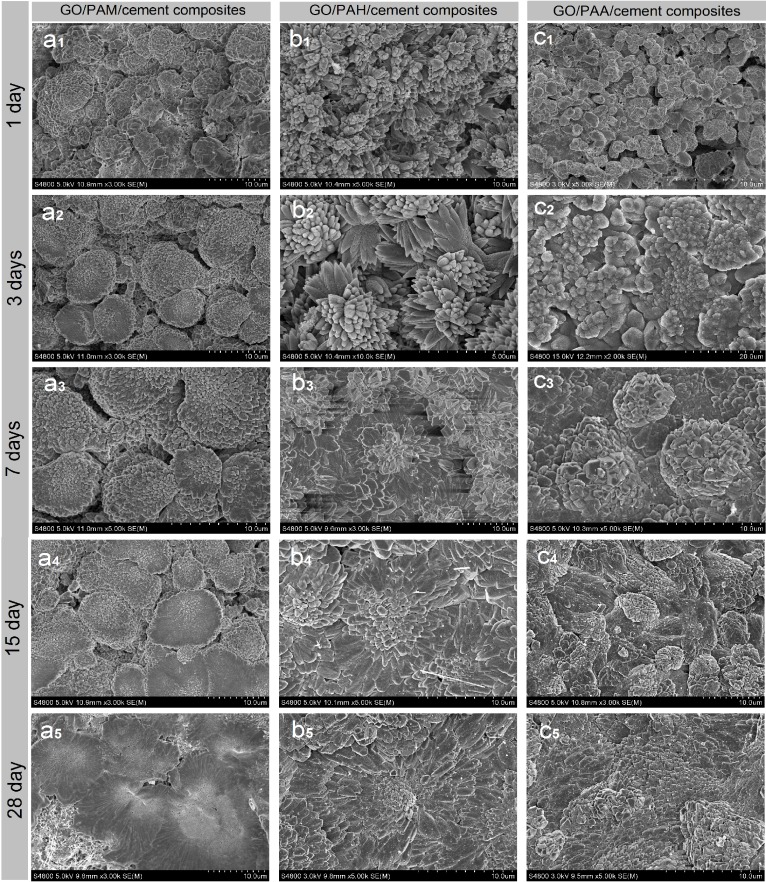
SEM images of GO/cement composites at different ages. (**a_1_**–**a_5_**): GO/PAM/cement composites; (**b_1_**–**b_5_**): GO/PAH/cement composites; (**c_1_**–**c_5_**): GO/PAA/cement composites.

**Figure 12 nanomaterials-08-00964-f012:**
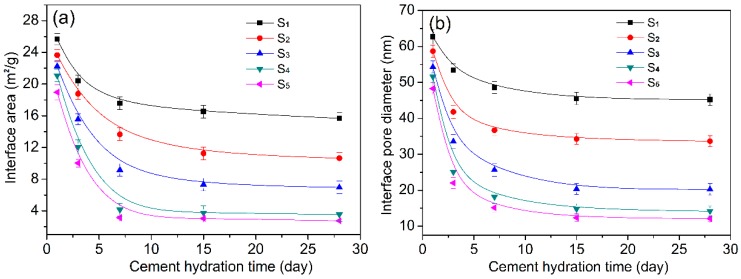
Interfacial areas (**a**) and pore diameters (**b**) within cement composites.

**Figure 13 nanomaterials-08-00964-f013:**
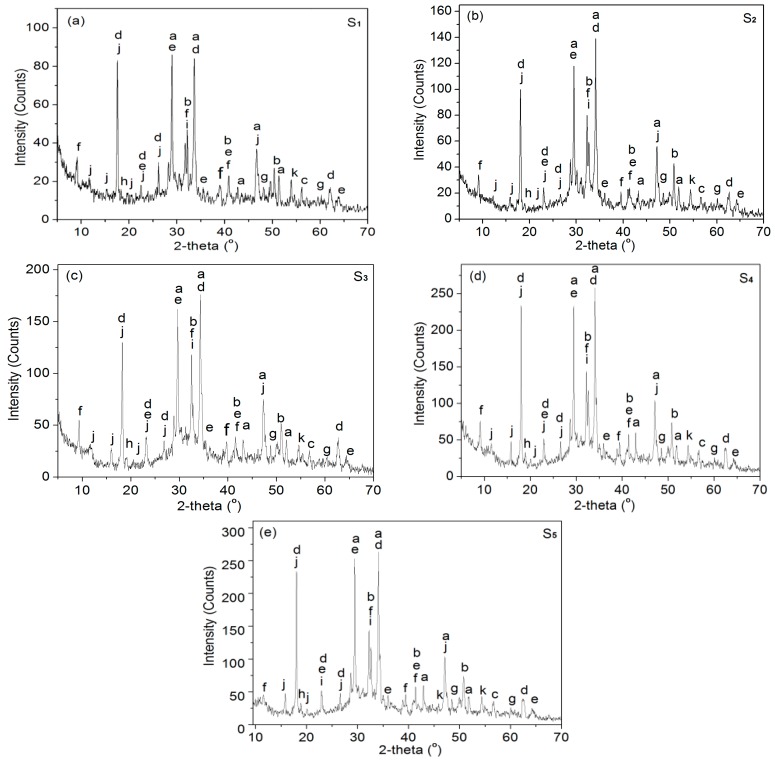
XRD patterns of cement composites at 28 day. (**a**) S_1_ sample; (**b**) S_2_ sample; (**c**) S_3_ sample; (**d**) S_4_ sample; (**e**) S_5_ sample. a: C_3_S, Ca_2_SiO_5_, Monoclinic; b: C_2_S, Ca_2_SiO_4_, Monoclinic; (c) SiO_2_, Tetragonal; d: CH, Ca(OH)_2_, Hexagonal; € CaCO_3_,Hexagonal; f: AFt, Ca_6_Al_2_(SO_4_)_3_(OH)_12_·26H_2_O, tetragonal; g: AFm,Ca_4_Al_2_O_6_(SO_4_)·14H_2_O, orthorhombic; h: C–S–H,Ca_3_Si_2_O_7_·xH_2_O, Amorphous; i: C–S–H, Ca_3_Si_2_O_7_·xH_2_O, Monoclinic; j: CaAl_2_Si_6_O_16_·6H_2_O, Tetragonal; k: CaHSi_2_O_7_, Orthorhombic.

**Table 1 nanomaterials-08-00964-t001:** *M*_w_, *M*_n_, and polydispersity (PDI) of polymer dispersants and their dielectric constant.

	Solid Content (%)	*M*_w_ (Da)	*M*_n_ (Da)	PDI	Dielectric Constant
PAM	20.36	57,251	38,736	1.48	12.54
PAH	20.52	54,258	35,475	1.53	32.65
PAA	20.44	41,895	30,586	1.37	43.54
PCs	20.38	53,687	37,245	1.44	11.35

**Table 2 nanomaterials-08-00964-t002:** Interfacial areas within cement composites.

	Interfacial Areas (m^2^/g)					Pore Diameter (nm)				
	S_1_	S_2_	S_3_	S_4_	S_5_	S_1_	S_2_	S_3_	S_4_	S_5_
1 day	25.68	23.65	22.23	21.08	18.89	62.68	58.65	54.26	51.58	48.27
3 day	20.41	18.78	15.56	12.06	10.03	53.41	41.78	33.56	25.06	22.03
7 day	17.57	13.67	9.15	4.15	3.14	48.57	36.67	28.65	18.15	15.14
15 day	16.52	11.25	7.32	4.74	3.05	45.52	34.25	20.35	14.74	12.34
28 day	15.68	10.65	6.96	4.25	2.92	45.18	33.65	20.23	14.17	12.12

**Table 3 nanomaterials-08-00964-t003:** Compressive strength of cement composites.

	Compressive Strength (MPa)					Increase Ratios (%)			
	S_1_	S_2_	S_3_	S_4_	S_5_	S_2_/S_1_	S_3_/S_1_	S_4_/S_1_	S_5_/S_1_
1 day	7.67 ± 0.41	8.42 ± 0.45	10.34 ± 0.48	11.34 ± 0.61	11.86 ± 0.63	9.78	34.81	47.85	54.63
3 day	28.21 ± 1.23	36.25 ± 1.45	40.75 ± 1.65	47.54 ± 1.84	55.67 ± 1.93	28.51	44.45	68.52	97.34
7 day	46.71 ± 1.76	61.72 ± 2.04	67.56 ± 2.14	73.34 ± 2.25	88.36 ± 2.45	32.13	44.64	57.01	89.17
15 day	50.33 ± 1.86	72.58 ± 2.18	82.62 ± 2.36	104.37 ± 2.57	112.73 ± 2.64	44.21	64.16	107.37	123.98
28 day	55.72 ± 1.87	78.31 ± 2.23	89.75 ± 2.38	116.82 ± 2.75	128.32 ± 2.81	40.54	61.07	109.66	130.29

**Table 4 nanomaterials-08-00964-t004:** Flexural strength of cement composites.

	Flexural Strength (MPa)					Increase Ratios (%)			
	S_1_	S_2_	S_3_	S_4_	S_5_	S_2_/S_1_	S_3_/S_1_	S_4_/S_1_	S_5_/S_1_
1 day	0.96 ± 0.35	1.26 ± 0.0.38	1.32 ± 0.39	1.56 ± 0.35	1.83 ± 0.36	31.25	37.50	62.50	90.63
3 day	2.23 ± 0.35	4.55 ± 0.42	5.26 ± 0.43	6.24 ± 0.48	7.33 ± 0.48	104.04	135.87	179.82	228.70
7 day	5.24 ± 0.43	7.27 ± 0.41	8.83 ± 0.46	10.42 ± 0.32	13.55 ± 0.41	38.74	68.51	98.86	158.59
15 day	6.86 ± 0.37	9.26 ± 0.38	10.65 ± 0.42	13.87 ± 0.43	16.67 ± 0.45	34.98	55.25	102.19	143.01
28 day	7.53 ± 0.38	10.85 ± 0.39	12.35 ± 0.42	15.97 ± 0.43	17.68 ± 0.44	44.09	64.01	112.09	134.79

**Table 5 nanomaterials-08-00964-t005:** Cement composite durability testing results.

	Penetration Resistance		Frost Resistance *			Carbonation Depth (mm)	
	Osmotic Pressure (MPa)	Seepage Height (mm)	*m*_0_ (g)	*m*_loss_ (g)	*p* (%)	7 day	28 day
S_1_	3.6	14.3	9843	0.15	73.6	3.2	4.6
S_2_	3.6	8.3	9831	0.08	88.5	1.7	1.69
S_3_	3.6	5.1	9837	0.01	94.8	0.3	0.6
S_4_	3.6	4.6	9856	0.01	95.7	0.3	0.5
S_5_	3.6	3.8	9678	0.01	96.8	0.3	0.4

* *m*_0_: Sample mass before freeze–thaw experiments. *m*_loss_: Sample mass after 100 freeze–thaw cycles. *p*: The retention rate of the relative dynamic elasticity modulus after 100 freeze–thaw cycles.

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
