# Peer review of "Investigation of the Effects of Polymer Dispersants on Dispersion of GO Nanosheets in Cement Composites and Relative Microstructures/Performances"

_nanomaterials, 2018, doi:10.3390/nano8120964_

Round 1

Reviewer 1 Report

There are some mistakes in the text:

- Line 191: correct shaeets to sheets.

- LIne 294:  Correct Figure 6 to figure 11.

The results are quite interesting and the authors have performed all the measurements needed to achieve the conclusions.

I have only a question: What about employing reduced graphene oxide, chemically or photochemically obtained, instead of GO? Maybe the dispersability is better than GO.

Author Response

Response to reviewers’ comments

Reviewer 1.

Open Review

English language and style

( ) Extensive editing of English language and style required
( ) Moderate English changes required
(x) English language and style are fine/minor spell check required
( ) I don't feel qualified to judge about the English language and style

√We have carefully checked and revised the English language and style.

Yes

Can be improved

Must be improved

Not applicable

Does the introduction   provide sufficient background and include all relevant references?

( )

(x)

( )

( )

Is the research design appropriate?

(x)

( )

( )

( )

Are the methods adequately   described?

(x)

( )

( )

( )

Are the results clearly   presented?

(x)

( )

( )

( )

Are the conclusions   supported by the results?

(x)

( )

( )

( )

  We have added some research background and relevant references in line 79-85.

  Comments and Suggestions for Authors

There are some mistakes in the text:

- Line 191: correct shaeets to sheets.

- LIne 294:  Correct Figure 6 to figure 11.

ü  We have modified these mistakes according to your advice.

The results are quite interesting and the authors have performed all the measurements needed to achieve the conclusions.

I have only a question: What about employing reduced graphene oxide, chemically or photochemically obtained, instead of GO? Maybe the dispersability is better than GO.

ü  Graphene oxide was prepared usually by Hummers’ method, that is, graphite was oxidized with concentrated sulfuric acid (H2SO4) and potassium permanganate (KMnO4). The graphene oxide have many oxygen-containing groups such as hydroxy, carboxy and epoxy groups. Therefore, graphene oxide have excellent hydrophilic and can disperse in water solution. With more oxygen-containing groups, the graphene oxide receive better dispersing in water.

The reduced graphene oxide was obtained by reducing the graphene oxide with some reducer. Therefore, the number of oxygen-containing groups is lower than the graphene oxide obtained directly from Hummers’ method without reducing with any reducer. The reduced graphene oxide would run far behind the graphene oxide in dispersing in water. In generally, the dispersability of reducing graphene oxide cannot cpmpare with graphene.            

Reviewer 2 Report

This interesting paper reports on the development of a new cement composite containing dispersed graphene oxide (GO) nanosheets. The new material shows lower porosity and improved mechanical properties. The Authors have used three polymer dispersants with different polar groups and dispersion capabilities, such as poly(acrylamide-methacrylic acid) (PAM), poly(acrylonitrile-hydroxyethyl acrylate) (PAH), and poly(allylamine-acrylamide) (PAA). The obtained composite materials have been thoroughly investigated using a range of techniques, including FTIR and XPS spectroscopy, XRD diffraction analysis, as well as the TEM, SEM, and AFM imaging. The Authors have shown that the distribution of GO nanosheets in cement composites induces the desired microstructure. The investigations of relationships between the interfacial area, pore size, and cement hydration time were also presented. The Authors have found that the compressive strength and durability indexes of the composites, including penetration, freeze-thaw, and carbonation of GO/polymer cement composites by GO intercalation were improved in comparison with blank samples.

The paper is of high interest to a broad Readership within graphene materials research groups and composite cement applications community. I recommend the paper for publication after minor revisions addressing the comments presented below.

1. When talking about "hydration crystals", do you mean "hydrated crystals"? Please check and correct appropriately.

2. The dispersant molecule (DM) net charge and also the local charges in these molecules are important to facilitate interactions with GO and prevent its re-stacking. The charge of the main polar groups, carboxyl and amine functionalities, is pH dependent; hence the discussion of their interactions with GO and interparticle repulsions between GO@DM should include the estimation of their charge in the relevant media. For instance, the NH2 groups are certainly protonated in solutions of neutral pH forming –NH3+ groups but are deprotonated in alkaline solutions. The electrostatic interactions between functionalized solid phase surfaces or nanoparticles have previously been analyzed for various systems (see, for instance: Biomaterials 32 (2011) 3312-3321; Biosensors and Bioelectronics 88 (2017) 114-121; Nanomaterials 8 (2018) 510; Biosensors and Bioelectronics 55 (2014) 379-385). It seems that PAA is positively charged and thus can interact strongly with negatively charged GO and then provide enough net positive charges to prevent GO restacking. On the other hand, PAM is likely to bear no net charge in neutral solutions, thus cannot prevent GO restacking to the same extent as PAA does. If exact pK values for dispersant molecules are not available, then at least a plausible explanation of molecular charges and their interactions with GO and between the GO@DM nanoparticles should be provided.

3. Another mode of DM binding to GO, namely the hydrogen binding, has not been taken into account by the Authors. It has been shown that for many functionalized nanoparticles, the modifying molecules can be attached by hydrogen binding, see for instance: J. Phys. Chem. B 121 (2017) 6822−6830. The interparticle hydrogen bonding may also be a factor in microstructure development (see, for instance: J. Phys Chem. B 119 (2015) 13227-13235). For GO@DM, such possibilities should also be mentioned in the paper and the relevant literature references listed above should be cited.

Author Response

Response to reviewers’ comments

Reviewer 2.

Open Review

English language and style

( ) Extensive editing of English language and style required
( ) Moderate English changes required
(x) English language and style are fine/minor spell check required
( ) I don't feel qualified to judge about the English language and style

√ We have also carefully checked the English language and style.

Yes

Can be improved

Must be improved

Not applicable

Does the introduction   provide sufficient background and include all relevant references?

( )

(x)

( )

( )

Is the research design   appropriate?

(x)

( )

( )

( )

Are the methods adequately   described?

(x)

( )

( )

( )

Are the results clearly   presented?

( )

(x)

( )

( )

Are the conclusions   supported by the results?

(x)

( )

( )

( )

  We have added some research background and relevant references in the introduction and references. We have added the explanation on the results such as dispersion mechanism of polymer dispersants for GO nanosheets. The added explanation is in line 230-247.

Comments and Suggestions for Authors

This interesting paper reports on the development of a new cement composite containing dispersed graphene oxide (GO) nanosheets. The new material shows lower porosity and improved mechanical properties. The Authors have used three polymer dispersants with different polar groups and dispersion capabilities, such as poly(acrylamide-methacrylic acid) (PAM), poly(acrylonitrile-hydroxyethyl acrylate) (PAH), and poly(allylamine-acrylamide) (PAA). The obtained composite materials have been thoroughly investigated using a range of techniques, including FTIR and XPS spectroscopy, XRD diffraction analysis, as well as the TEM, SEM, and AFM imaging. The Authors have shown that the distribution of GO nanosheets in cement composites induces the desired microstructure. The investigations of relationships between the interfacial area, pore size, and cement hydration time were also presented. The Authors have found that the compressive strength and durability indexes of the composites, including penetration, freeze-thaw, and carbonation of GO/polymer cement composites by GO intercalation were improved in comparison with blank samples.

The paper is of high interest to a broad Readership within graphene materials research groups and composite cement applications community. I recommend the paper for publication after minor revisions addressing the comments presented below.

1.    When talking about "hydration crystals", do you mean "hydrated crystals"? Please check and correct appropriately.

ü When cement meet water, it will carry out chemical reaction. The reaction is also called as hydrating reaction, and the reaction products in the hydrating processes are called hydration products. If the reaction products are crystals, the hydration products is also hydration crystals.

2. The dispersant molecule (DM) net charge and also the local charges in these molecules are important to facilitate interactions with GO and prevent its re-stacking. The charge of the main polar groups, carboxyl and amine functionalities, is pH dependent; hence the discussion of their interactions with GO and interparticle repulsions between GO@DM should include the estimation of their charge in the relevant media. For instance, the NH2 groups are certainly protonated in solutions of neutral pH forming –NH3+ groups but are deprotonated in alkaline solutions. The electrostatic interactions between functionalized solid phase surfaces or nanoparticles have previously been analyzed for various systems (see, for instance: Biomaterials 32 (2011) 3312-3321; Biosensors and Bioelectronics 88 (2017) 114-121; Nanomaterials 8 (2018) 510; Biosensors and Bioelectronics 55 (2014) 379-385). It seems that PAA is positively charged and thus can interact strongly with negatively charged GO and then provide enough net positive charges to prevent GO restacking. On the other hand, PAM is likely to bear no net charge in neutral solutions, thus cannot prevent GO restacking to the same extent as PAA does. If exact pK values for dispersant molecules are not available, then at least a plausible explanation of molecular charges and their interactions with GO and between the GO@DM nanoparticles should be provided.

ü We have added the explanation on the dispersing mechanism of dispersants of PAA, PAH and PAM. Especially the dispersing mechanism of PAA for GO nanosheets has been give more explanation. The explanation was added in Line 230-247. The recommended references have been added in the references of the manuscript.

3. Another mode of DM binding to GO, namely the hydrogen binding, has not been taken into account by the Authors. It has been shown that for many functionalized nanoparticles, the modifying molecules can be attached by hydrogen binding, see for instance: J. Phys. Chem. B 121 (2017) 6822−6830. The interparticle hydrogen bonding may also be a factor in microstructure development (see, for instance: J. Phys Chem. B 119 (2015) 13227-13235). For GO@DM, such possibilities should also be mentioned in the paper and the relevant literature references listed above should be cited.

ü We have added the explanation about hydrogen bonding and Van der Waals force of dispersants and GO nanosheets in line 230-249. The recommended have been cited in the manuscript.

Reviewer 3 Report

Authors describe a methodology to prepare a new type of cement composites by modification with polymer-GO conjugates nanomaterials.

They performed the synthesis of GO, preparation of different polymers, conjugates and finally GO-P-cement mixtures. They performed a well characterization of the different materials synthesized.

However, the polymer modification of the GO must be clarify. This point is of high relevant in the final work because polymer seems to have important role in the final cement composite preparation.

Three different polymers were used to interact with GO although no explanation or discussion about which is the chemical interaction between them has been performed. PAA is the clearest example where the ionic interaction between amino from polymer and carboxylate from GO seems to be the chemical interaction.

But how is with PAM or PAH??? Are there some covalent binding??

Why was not tried the covalent attachment as methodology by introducing carbodiimide in the reaction conditions????

Better uniform sheets were formed using PAA polymer, this seems to be related with the chemical interaction between both compounds.

the size range of GO was so broad in the case of use PAM and PAH, which is the explanation???

Fig 6 is completely unclear, and this must be better performed considering exactly how the chemical reaction is in each case. Also this strategy must be better explain in the manuscript.

Which is the exact composition of the cement??? How is the interaction between cement and GO-polymer conjugates???

Conclusion must be rewritten, emphasizing the advantages of the modification of cements with GO in the material properties, instead of a list of results described previously. For example, about the flexural strengths, it was more than 3 times stronger in which case?? Because there a list of number without explanation. Emphasize the best results obtained with the final objective.

Author Response

Response to reviewers’ comments

Reviewer 3

Open Review

English language and style

( ) Extensive editing of English language and style required
( ) Moderate English changes required
(x) English language and style are fine/minor spell check required
( ) I don't feel qualified to judge about the English language and style

√ We have carefully   checked the English language and style.

Yes

Can be improved

Must be improved

Not applicable

Does the introduction   provide sufficient background and include all relevant references?

(x)

( )

( )

( )

Is the research design   appropriate?

( )

( )

(x)

( )

Are the methods adequately   described?

( )

( )

(x)

( )

Are the results clearly   presented?

( )

( )

(x)

( )

Are the conclusions   supported by the results?

( )

( )

(x)

( )

  We have rechected the description about the experiment methods.

   We have revised the results and discussion for clearly understand. The conclusion have been rewritten.

Comments and Suggestions for Authors

Authors describe a methodology to prepare a new type of cement composites by modification with polymer-GO conjugates nanomaterials.

They performed the synthesis of GO, preparation of different polymers, conjugates and finally GO-P-cement mixtures. They performed a well characterization of the different materials synthesized.

However, the polymer modification of the GO must be clarify. This point is of high relevant in the final work because polymer seems to have important role in the final cement composite preparation.

Three different polymers were used to interact with GO although no explanation or discussion about which is the chemical interaction between them has been performed. PAA is the clearest example where the ionic interaction between amino from polymer and carboxylate from GO seems to be the chemical interaction.

But how is with PAM or PAH??? Are there some covalent binding??

ü The intercalation between dispersants and GO nanosheets have been explained in lines 230-249.

Why was not tried the covalent attachment as methodology by introducing carbodiimide in the reaction conditions????

ü In this paper, our aim is to investigate the effects of functional groups in the polymer dispersants on dispersability of GO nanosheets. So we designed and synthesized three polymer dispersants with different polar groups (weak, mild, and strong)  for investigating their ability to disperse GO nanosheets. The polymer dispersants poly(acrylamide-methacrylic acid) (PAM), poly(acrylonitrile-hydroxyethyl acrylate) (PAH), and poly(allylamine-acrylamide) (PAA) were synthesized via free-radical copolymerization with a monomer molar ratio of 1:3 according to order of polymer name.

Better uniform sheets were formed using PAA polymer, this seems to be related with the chemical interaction between both compounds.

Te size range of GO was so broad in the case of use PAM and PAH, which is the explanation???

ü The size range of GO was broader distribution using PAM and PAH dispersants compared with using PAA dispersant. The reason is that GO is multilayered agglomeration in the case of using PAM and PAH, and the agglomeration of GO nanosheets have uncertainly volumes and shapes. So the size range of GO tested by using a NANO-ZS90 laser particle analyzer are broad. By Contrast, the size range of GO by using PAA as dispersant have small and narrow distribution, the reason is that the GO nanosheets are few-layered (1-2 layers) dispersion. So the single GO nanosheets are small and its size range is narrow.

Fig 6 is completely unclear, and this must be better performed considering exactly how the chemical reaction is in each case. Also this strategy must be better explain in the manuscript.

ü The figure 6 shows the schematic diagram of dispersing mechanism of GO nanosheets with different dispersants. The schematic diagram of dispersing mechanism are proposed based on the results of the size distribution, AFM images and TEM images of GO nanosheets. The explanation of Figure 6 was listed in line 230-250.

Which is the exact composition of the cement??? How is the interaction between cement and GO-polymer conjugates???

ü Cement in a hydrous state mainly consists of tricalcium silicate C3S (Ca3SiO5), dicalcium silicate C2S (Ca2SiO4), tricalcium aluminate C3A (Ca3Al2O6), tetracalcium aluminoferrite C4AF (Ca4AlnFe2-nO7) as well as a small amount of clinker sulfate (Na2SO4, Ka2SO4) and gypsum (CaSO4·2H2O). In the hydration process, C3A, C4AF, C3S and C2S will carry out a complex hydration reaction to form ettringite (Ca6Al2(SO4)3)(OH)12·26H2O, AFt), Ca4Al2(OH)2·SO4·H2O, AFm), calcium hydroxide (Ca(OH)2, CH) and calcium silicate hydrate (3CaO·2SiO2·4H2O, C–S–H) gel, respectively. Generally, CH, AFt and AFm exhibit rod-like and needle-like shapes with disorder, which determines the brittleness of cement paste.

According to the research result, we think the role of GO is to control the shapes and aggregate state of cement reaction products. The role of the polymer dispersant is only to diperse the GO into few-layered dispersion and uniformly distributing in cement composites. The reason is that GO has many oxygen functional groups main included of –OH, COOH and –SO3H. The active functional groups react preferentially with C3S, C2S and C3A and form the growth points of the hydration products. The growth points and growth pattern of the hydration products are both controlled by GO, which is called a template effect. GO can make many neighboring rod-like hydration crystals on the same GO surface form a thick column-like shape and flower shaped crystals. These columnar products consist of rod-like of AFt, AFm, CH and CS–H and grow forward from the GO surface in the same direction due to great stress around them, keeping the column shape. Once the column-shaped crystals grow into a pore, crack or loose structure, they grow apart and form the fully-bloomed flower-like crystals, which disperse in pores and cracks as fillers and crack arrestors to retard crack propagation. The flower-like crystals usually generated in holes and gap of the cement composites and forming cross-linking structure have great contributed to improving toughness of cement composites.

The explanation have listed in the previously research paper. So the content of interaction between cement and GO-polymer conjugates may see in previously paper.

Conclusion must be rewritten, emphasizing the advantages of the modification of cements with GO in the material properties, instead of a list of results described previously. For example, about the flexural strengths, it was more than 3 times stronger in which case?? Because there a list of number without explanation. Emphasize the best results obtained with the final objective.

ü We have rewritten conclusion according your advice. Now, the conclusion the results obtained by these research project.

Reviewer 4 Report

Let me first state that I have found the document interesting and very well structured. The information presented supports the conclusions of the work, and it may be useful for the community, so I will recommend it for publication once some changes / comments are included.

This is a minor point, but I think that, while the rest of the paper is very well written, the Introduction has several typos and some sentences that should be revised. Some examples: (i) Sentence in lines 49-50 is not finished (or better said, it is finished with a dot while it should be continued with a comma. (ii) line 58 "this problems", (iii) Line 74-75 "PCs cannot... due to its poor..." (instead of their) Line 76: the sentence beginning with "While" is not finished or should be linked to the previous one. Line 85: "Intercaltion"....  I strongly recommend to carefully check the grammar and syntax of this section. I have found less mistakes/typos in the rest of the document. In line 294 "Figure6" is not separated, There is a typo in Section 3.4. name. To the best of my knowledge, "include of" is not correct. Line 191: shaeets. Please check the whole text.

There is a very important reference that the authors did not include, and this is a little surprising as it is their own work in Nanomaterials, Dec. 2017. That work is relevant and should indeed be not only cited, but compared with in my opinion. It is clear that the approach of both works are similar, although different materials were considered in each case. Can the conclusions of both works be compared? Please comment.

How is the Interface area (Figure 12, Table 2) measured? The authors comment in Methods Section on the Interface Pore diameter measurement: can they both be extracted from the AutoPore measurements?

I don't understand well how Figure 6 is obtained. Is it qualitative? Or is it supported by some numerical calculations giving the lower energy states of the composites? (For example, DFT or similar, simpler calculations). Please comment on this.

In Figures 7...10, please make note in the figure foot on the scales for each figure.

Figure 13 is, in my opinion, difficult to follow. The 3D view complicates the comparison of the peaks in each of the samples. I guess a standard 2D figure does not make it simpler, or does it? I would really like to see it...

For those who are not expert in durability analysys, the penetration parameters may not be as simple and I would recommend to include at least a citation when presenting them (lines 403-404). Regarding these, Table 5 depicts some 0.0 results for m_loss. Does that mean that the mass loss is lower than the end of the scale of measurement? Shouldn't the data include, then, the measurement error? 

Again, let me stress I have really enjoyed reading and reviewing this work. I think that with some small changes it could deserve publication in Nanomaterials.

Author Response

Response to reviewer’s comments

Reviewer 4

Open Review

English language and style

( ) Extensive editing of English language and style required
(x) Moderate English changes required
( ) English language and style are fine/minor spell check required
( ) I don't feel qualified to judge about the English language and style

√ We have also carefully checked the English language and style.

Yes

Can be improved

Must be improved

Not applicable

Does the introduction   provide sufficient background and include all relevant references?

( )

(x)

( )

( )

Is the research design   appropriate?

(x)

( )

( )

( )

Are the methods adequately   described?

(x)

( )

( )

( )

Are the results clearly   presented?

(x)

( )

( )

( )

Are the conclusions   supported by the results?

(x)

( )

( )

( )

  We have added some research background and relevant references.

Comments and Suggestions for Authors

Let me first state that I have found the document interesting and very well structured. The information presented supports the conclusions of the work, and it may be useful for the community, so I will recommend it for publication once some changes / comments are included.

This is a minor point, but I think that, while the rest of the paper is very well written, the Introduction has several typos and some sentences that should be revised. Some examples:

(i) Sentence in lines 49-50 is not finished (or better said, it is finished with a dot while it should be continued with a comma.

(ii) line 58 "this problems",

(iii) Line 74-75 "PCs cannot... due to its poor..." (instead of their). Line 76: the sentence beginning with "While" is not finished or should be linked to the previous one.

Line 85: "Intercaltion"....  I strongly recommend to carefully check the grammar and syntax of this section. I have found less mistakes/typos in the rest of the document.

In line 294 "Figure6" is not separated, There is a typo in Section 3.4. name. To the best of my knowledge, "include of" is not correct.

Line 191: shaeets. Please check the whole text.

ü We have revised these mistakes in the revision according to your guidance. The corrected words or sentences marked with red.

There is a very important reference that the authors did not include, and this is a little surprising as it is their own work in Nanomaterials, Dec. 2017. That work is relevant and should indeed be not only cited, but compared with in my opinion. It is clear that the approach of both works are similar, although different materials were considered in each case. Can the conclusions of both works be compared? Please comment.

ü  We have add the content about “Nanomaterials, 2017,7, 457”in “Introduction” of Line 73-86. We also give an explanation about the different between the manuscript and “Nanomaterials, 2017,7, 457”.

For carboxymethyl chitosan, it has excellent dispersability for GO nanosheets and can disperse the multi-layered GO agglomeration into few-layered GO nanosheets (1–2 layers) due to its amphoteric polymeric structure, but it has high price and is not appropriate widespread use. However, the amphoteric carboxymethyl chitosan enlightens us to synthesize polymeric amphoteric dispersant for replacing carboxymethyl chitosan.

How is the Interface area (Figure 12, Table 2) measured? The authors comment in Methods Section on the Interface Pore diameter measurement: can they both be extracted from the AutoPore measurements?

ü  In this researchthe interface area and interface pore diameters were measured using an AutoPore IV9500 automatic mercury porosimeter (Norcross, GA, USA). The instrument by its self cannot know the shapes of pores. We usually can obtain pore structure such as pore area, pore diameter distribution and average pore diameter using this instrument, but we don’t know what the pores’ shapes looks like by these testing data. In generallythe pores shapes may be know by observing their microstructure with SEM or TEM images.

   In this manuscript, we fist tested and analyzed the formation process of producing cement hydration products to forming regularly shaped crystals that aggregate into compact microstructures at 28 d, and the results are shown in Figure 8-11 and the explanation are shown in Section 3.2 and 3.3. The major conclusions are that “GO nanosheet-induced formation of ordered microstructures takes place gradually. The outstanding structural feature is that there is greater porosity in the early days of hydration ages (1-7d). The porosity includes pores and cracks that form are mostly interfacial gaps between crystalline reaction products. These interfacial gaps may be characterized by measuring the porosities of cement composites ” by using an AutoPore IV9500 automatic mercury porosimeter

I don't understand well how Figure 6 is obtained. Is it qualitative? Or is it supported by some numerical calculations giving the lower energy states of the composites? (For example, DFT or similar, simpler calculations). Please comment on this.

ü The figure 6 shows the schematic diagram of dispersing mechanism of GO nanosheets with different dispersants. The schematic diagram of dispersing mechanism are proposed based on the results of the size distribution, AFM images and TEM images of GO nanosheets.

In Figures 7...10, please make note in the figure foot on the scales for each figure.

ü We have added the size mark on the Figure 7-10.

Figure 13 is, in my opinion, difficult to follow. The 3D view complicates the comparison of the peaks in each of the samples. I guess a standard 2D figure does not make it simpler, or does it? I would really like to see it...

ü  In figure 13the five XRD patterns are shown in the 3D figure and can clearly show their differences of XRD patterns. Accordingly, the differences of cement hydration products in different samples can easily know.

For those who are not expert in durability analysys, the penetration parameters may not be as simple and I would recommend to include at least a citation when presenting them (lines 403-404). Regarding these, Table 5 depicts some 0.0 results for m_loss. Does that mean that the mass loss is lower than the end of the scale of measurement? Shouldn't the data include, then, the measurement error? 

ü The results indicated that the cement composites with GO nanosheets have excellent volume stability, and its test samples did not weight loss after the testing process. 

Again, let me stress I have really enjoyed reading and reviewing this work. I think that with some small changes it could deserve publication in Nanomaterials.

Round 2

Reviewer 3 Report

Authors improved the manuscript in this new revised version. However, Fig 6 must be improved. The quality of the scheme is quite poor, specially the GO.

New high quality figure must be performed including clearly the chemical groups in GO and the interactions between these groups and polymer groups.

Author Response

Comments and Suggestions for Authors Authors improved the manuscript in this new revised version. However, Fig 6 must be improved. The quality of the scheme is quite poor, specially the GO. New high quality figure must be performed including clearly the chemical groups in GO and the interactions between these groups and polymer groups. √ We have changed the Figure 6 according to your advice. Please see Figure 6 and its description in line 225-243. We also checked carefully the manuscript.

Reviewer 4 Report

The authors have answered to (almot all of) my queries. Thank you very much for the clarifications, I find this new version of the work suitable for publication. 

I would only mark a couple of (very minor) points:

- I still have detected a few minor English mistakes, so I recommend to double check (for example, the second conclusion says "The microstructure are") 

- Also, while Figure 13 clearly shows the differences, I still think a 2D figure would be more useful for a quantitative analysis/comparison. Nevertheless, let's agree we disagree in this point, I don't find it necessary for the paper purposes.

- Finally, while I agree on the conclusion regarding m_loss, my point (which I think was maybe not well understood) was that a 0.0 measurement does not mean a 0.0 weight loss if the end of the scale of the instrument is 0.0. As the error of the measurements is not included in this last table, one may wonder whether the end of the scale of the instrument is actually 0.00 grams or lower (and it was truncated). Nevertheless, again, the conclusion is well supported: I just couldn't avoid my Bachelor in Physics Prof's voice coming into my review...  

Nice work.

Author Response

Comments and Suggestions for Authors

The authors have answered to (almot all of) my queries. Thank you very much for the clarifications, I find this new version of the work suitable for publication. 

I would only mark a couple of (very minor) points:

- I still have detected a few minor English mistakes, so I recommend to double check (for example, the second conclusion says "The microstructure are") 

We have revised the mistake. We also checked carefully the manuscript.

- Also, while Figure 13 clearly shows the differences, I still think a 2D figure would be more useful for a quantitative analysis/comparison. Nevertheless, let's agree we disagree in this point, I don't find it necessary for the paper purposes.

We have changed the Figure 13 according to your advice. Please see Figure 13 and its description in line 380-393.

- Finally, while I agree on the conclusion regarding m_loss, my point (which I think was maybe not well understood) was that a 0.0 measurement does not mean a 0.0 weight loss if the end of the scale of the instrument is 0.0. As the error of the measurements is not included in this last table, one may wonder whether the end of the scale of the instrument is actually 0.00 grams or lower (and it was truncated). Nevertheless, again, the conclusion is well supported: I just couldn't avoid my Bachelor in Physics Prof's voice coming into my review...  

√ We have revised the mloss of S3, S4 and S5 in Table 5 according to the experiment results. The 0.01g of mloss just mean the freeze-thaw mass loss is very lower than other samples.
